# Flatness Has a Shape: Scalar Curvature and Functional Dimension in Neural Loss Landscapes

Johanna Marie Gegenfurtner[*1]

[1]Technical University of Denmark
{johge@dtu.dk}

## 1 Introduction

What does it mean for a loss surface to be flat? A *flat* minimum is one where the loss increases slowly in many directions around the optimum. Intuitively, a flat basin gives room for parameter perturbations without harming performance, which suggests robustness to noise, and potentially better generalisation. A common quantitative measure is the trace of the Hessian, which is the sum of its eigenvalues [1, 2]. Intuitively, large eigenvalues correspond to steep curvature in some directions. Hence, penalising or bounding those helps in finding flatter minima. Alternatively, the scalar curvature of the loss has been suggested [3] as a measure of flatness, carrying a more geometric flavour. Instead of only measuring individual directions, it combines curvature across two-dimensional planes (sectional curvatures) into a scalar at each point in the parameter space. In that sense, it is more geometrically meaningful and possibly more robust to coordinate changes. In this work, we will derive a novel bound of the scalar curvature in terms of the functional dimension and the eigenvalues, which is stated in Theorem 4.1. For coherence with related work, we only consider architectures with ReLU activations. The results can easily be stated more generally though.

**Notation** Assume we are given data points $\{x_i, y_i\}_{i=1}^N$, where $x_i \in \mathcal{X} \subseteq \mathbb{R}^d$ and $y_i \in \mathcal{Y} \subseteq \mathbb{R}^D$. In machine learning, our goal is to find a function

$$f_\theta : \mathcal{X} \to \mathcal{Y},$$

which best fits the data points and provides reasonable predictions for new points from the same distribution. We consider multi-layer perceptrons (MLPs) of $L$ layers with ReLU activation functions, parametrised by $\theta = \{W_j, b_j\}_{j=1,\dots L} \subseteq \Theta$. Let $f : \Theta \times \mathbb{R}^d \to \mathbb{R}^D$, be the layer wise composition

$$f_\theta(x) = w_L \sigma(w_{L-1}(\dots \sigma(w_1 x + b_1) + b_{L-1})) + b_L.$$

Here, $\sigma(z) = \max(0, z)$, and in each layer

$$w_j \in \mathbb{R}^{d_j \times d_{j-1}}, b_j \in \mathbb{R}^{d_j},$$

where $d_L = D$, and $d_0 = d$. To evaluate which function fits best, we define the empirical loss function

$\mathcal{L} : \Theta \to \mathbb{R}$. The mean squared error (MSE), for example, is given by

$$\mathcal{L}(\theta) = \frac{1}{2} \cdot \sum_{i=1}^N ||f_\theta(x_i) - y_i||^2.$$

## 2 The curvature of the loss manifold

We consider the loss manifold, which is the graph of the loss function:

$$\mathcal{M} = (\Theta, \mathcal{L}(\Theta)).$$

It can be equipped with the pull back metric

$$g(\theta) = I + \nabla_\theta \mathcal{L}^t \nabla_\theta \mathcal{L},$$

which equips each point of $\mathcal{M}$ with a scalar product, and hence a way to measure distances on the manifold. Using the metric, we can also evaluate the scalar curvature of $\mathcal{M}$.

A more comprehensive introduction to the required tools of differential geometry will be provided in a future version of this work. For now, we refer the interested reader to the classic textbook [4].

**Theorem 2.1.** *[3] The scalar curvature of $\mathcal{M}$ is given by*

$$\begin{aligned} K(\theta) &= \beta(\mathbf{trace}(H)^2 - \mathbf{trace}(H^2)) \\ &+ 2\beta^2 (\nabla_\theta \mathcal{L}(\theta)^t (H^2 - \mathbf{trace}(H)H) \nabla_\theta \mathcal{L}(\theta)), \end{aligned}$$

*where $H = \nabla_\theta^2 \mathcal{L}(\theta)$, and $\beta = \left(1 + ||\nabla_\theta \mathcal{L}(\theta)||^2\right)^{-1}$.*

We will call a point $\theta^*$ in the parameter space an interpolation solution if $f_{\theta^*}$ fits the training points perfectly, i.e. for all $i = 1, \dots, N$ we have $f_{\theta^*}(x_i) = y_i$. In this case, it is easy to verify that $\beta = 1$ and $\nabla_\theta \mathcal{L} = 0$, and hence the second summand of $K(\theta^*)$ vanishes.

## 3 Functional dimension

We will now define the functional dimension of a set of parameters. It measures the number of independent directions in the parameter space that change the function $f_\theta$.

---

[*]Corresponding Author.

We first give a definition depending on the choice of evaluation points. In the following we will assume that they are *parametrically smooth*, i.e. they lie in regions where the gradient of the function is well defined. This is a realistic assumption, since the complement of those regions has measure zero.

**Definition 3.1.** [5] Let $Z = \{x_1, \ldots, x_n\}$ be a set of points in $\mathbb{R}^d$, and

$$J_\theta = \begin{bmatrix} \nabla_\theta f_\theta(x_1) \\ \vdots \\ \nabla_\theta f_\theta(x_n) \end{bmatrix}.$$

We define the batch functional rank of $f_\theta$ as $\mathbf{rank}(J_Z(\theta))$.

Taking the supremum over all possible finite subsets $Z$, we obtain a more general definition.

**Definition 3.2.** [6] The functional dimension of $\theta \in \Theta$ is given by

$$\mathbf{dim}_{\text{fun}}(\theta) = \sup_Z \{\mathbf{rank}(J_Z(\theta)) \mid Z \subseteq \mathbb{R}^d \},$$

where all $Z$ are finite sets of parametrically smooth points for $\theta$.

We see that for a sufficiently large and spread out data set $Z$, the batch functional rank over $Z$ approximates the functional dimension of a function $f_\theta$, i.e.

$$\mathbf{rank}(J_Z(\theta)) \approx \mathbf{dim}_{\text{fun}}(\theta).$$

# 4 A new bound of the scalar curvature

A simple computation shows that

$$\nabla_\theta^2 \mathcal{L}(\theta) = \frac{1}{n} J_\theta^t J_\theta.$$

Hence, $\mathbf{rank}(J_\theta) = \mathbf{rank}(J_\theta^t J_\theta)$ equals the number of non-zero eigenvalues of $\nabla_\theta^2 L(\theta)$.

This yields another bound for the scalar curvature:

**Theorem 4.1.** *Let $\lambda_{max}$ denote the largest eigenvalue of $\nabla_\theta^2 \mathcal{L}(\theta)$, and $r = \mathbf{rank}(J_Z(\theta))$ be the batch functional rank of $f_\theta$ over the training points $Z = \{x_i\}$. The curvature $K(\theta^*)$ at an interpolation solution is bounded as follows:*

$$0 \leq K(\theta^*) \leq (r-1)\mathbf{trace}\left(\nabla_\theta^2 L(\theta^*)^2\right) \leq (r-1)\lambda_{max}^2.$$

*Proof.* From Theorem 2.1, we have that

$$K(\theta^*) = \mathbf{trace}(H)^2 - \mathbf{trace}(H^2)$$

$$= \left(\sum_i \lambda_i\right)^2 - \sum_i \lambda_i^2$$

$$\leq r \cdot \sum_i \lambda_i^2 - \sum_i \lambda_i^2$$

$$= (r-1) \cdot \sum_i \lambda_i^2.$$

Here we leveraged the Cauchy-Bunyakovski-Schwarz inequality, which implies

$$\left(\sum_i \lambda_i\right)^2 \leq r \cdot \sum_i \lambda_i^2.$$

$\square$

This shows that both the functional dimension and the eigenvalues of the Hessian bound the scalar curvature.

# 5 Future directions

**The precise relationship between functional dimensions and flatness.** A lower functional dimension implies that there are more flat directions and a higher-dimensional space of symmetries, as conjectured in [6]. Intuitively, this should correlate with local flatness. However, it is not trivial to show under which conditions this can be shown algebraically. For example, it might be the case that $\mathbf{dim}_{\text{fun}}(\theta) = 1$, but the corresponding eigenvalue is very large, which implies a sharp minimum. Conversely, for a large functional dimension we can still obtain a flat minimum if all eigenvalues are small. The aim of our work is to how we can bound both quantities, and how they control the scalar curvature.

**Parameter space symmetries.** As mentioned above, symmetries of the parameter space induce flat directions. They depend on the chosen architecture, but have been extensively studied for ReLU architectures [7–9]. Can we leverage this information to say something about the scalar curvature?

**Empirical studies.** What remains to work on is an empirical study of the sectional curvature, eigenvalues and functional dimension during training. In [6], it has been conjectured that a lower functional dimension corresponds to local flatness of the loss. This could yield insights into implicit regularisation of stochastic gradient descent (SGD). As demonstrated in [10], SGD implicitly regularises the terms $||\nabla_\theta f_\theta(x_i)||$. We ask if SGD (or other training methods) also control functional dimension.

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
