# OpenReview forum: "Flatness Has a Shape: Scalar Curvature and Functional Dimension in Neural Loss Landscapes"
_NLDL.org/2026/Abstracts_Track — NLDL 2026 Abstracts_

### Official Review · Reviewer_tWbQ · 2025-11-01

**Soundness:** 3
**Correctness:** 4
**Rating:** 4
**Confidence:** 3

**Summary:**

The paper investigates the geometry of neural loss landscapes using scalar curvature as a measure of flatness. The author derives a bound that links scalar curvature, the functional dimension and the eigenvalues of the Hessian, thereby connecting geometric and functional perspectives on flatness. Scalar curvature, which aggregates curvature across all two-dimensional planes (sectional curvatures), offers a coordinate-independent and geometrically meaningful description of the loss surface. The main finding shows that curvature is governed jointly by the effective number of active parameter directions—the functional dimension—and the Hessian’s spectrum.

**Strengths:**

**TL;DR**: Elegant theorem linking scalar curvature to functional dimension and Hessian eigenvalues, providing a geometric framework for understanding flat minima in neural networks. This has high potential for further discussions and extensions.

**Long Version**:
- The paper provides a clear and mathematically well-formulated theorem connecting curvature, the Hessian spectrum, and functional dimension—an elegant synthesis of geometric and analytic approaches.
- Although the result is relatively compact, it represents a meaningful conceptual step forward toward a geometric understanding of flat minima and generalization in neural networks.
- The extensive discussion of open questions and future directions is likely to stimulate further research in the intersection of geometric deep learning and optimization theory.

The paper sets a proper foundation for further discussions specifically when it comes to practical applications and implications for training and interpreting neural networks.

**Weaknesses:**

**TL;DR**: Lacks clear practical implications, missing abstract, and contains verbose notation sections that could be streamlined for the target audience.

**Long Version**:
- (major) The paper does not clearly articulate the implications of the derived bound for real training dynamics or model generalization. While the “Future directions” section alludes to potential empirical studies, an explicit statement connecting the theoretical bound to practical flatness measures or optimization strategies would strengthen the paper. What does bounding curvature by functional rank *tell us* geometrically or statistically about neural networks?
- (major) No discussion of computational complexity. Computing scalar curvature requires full Hessian; functional dimension requires Jacobian rank over potentially large datasets. Are these practical to measure compared to related works?
- (minor) It is unusual for a submission to omit an abstract; even a short summary would help readers quickly grasp the motivation and main result.
- (minor) Missing discussion of why scalar curvature is the "right" notion of flatness. The paper asserts it's "more geometrically meaningful" but doesn't compare it to other flatness measures (sharpness, PAC-Bayes bounds, etc.) theoretically or empirically. This might extend beyond the scope of an extended abstract, so this point is 'minor' here.
- (major) The supremum definition of functional dimension (Definition 3.2) is impractical to compute. The paper doesn't discuss convergence: how large must Z be before rank(J_Z) ≈ dim_fun?
- (minor) The section introducing MLP notation could be substantially condensed, as the audience for loss landscape research can safely be assumed familiar with such basics. This would free space for deeper discussion of the implications of the result.
- (minor) The text could clarify what is meant by the manifold being the *graph* of the loss function—whether this is intended as a formal embedding of the loss surface in parameter–loss space or a discrete set of evaluation points. A single explanatory sentence would resolve this ambiguity.
- (minor) The bibliography lacks consistency in formatting (URLs, DOIs, citation style).

---

### Official Review · Reviewer_mrdh · 2025-11-03

**Soundness:** 4
**Correctness:** 4
**Rating:** 5
**Confidence:** 3

**Summary:**

In this work, a bound is derived for the scalar curvature of the loss manifold, which is related to the gradients of the loss. The derivation uses a notion of functional dimension, which is related to rank of the matrix involving the gradient of the mapping function. This is then related to the loss gradient, and subsequently leads to the derivation of the bound, building upon a result in [3].

**Strengths:**

The work involves a clear, well defined, and rigorous contribution
It is also well written and with a reasonably good flow connecting the concepts of scalar curvature, functional dimension, and the eventual derivation of bound.
Finally the future extensions and practical implications of the work are discussed.

**Weaknesses:**

Some questions to clarify

1) The purpose of defining pull-back metric is not clear, as it is not used in the rest of the abstract.

2) Since the functional dimension is defined based on a set of points, does the no. of points affect the rank. Does it imply that one would get a consistent rank (invariant to the no. of points), given a sufficiently large number of points?

3) What are the computational implications of question 2, above?

---

### Official Review · Reviewer_HXDM · 2025-11-03

**Soundness:** 2
**Correctness:** 3
**Rating:** 4
**Confidence:** 3

**Summary:**

A deeper analysis of loss functions, looking at their shape rather than just their values.

**Strengths:**

See comment on weaknesses

**Weaknesses:**

Rather unclear which research gap the work is addressing, and how this differs from already done work using the rate of change.

---

### Decision · Program_Chairs · 2025-11-05

**Decision:**

Accept

**Comment:**

The abstract is of interest to the community and should be presented at the conference.